# Analysis of Early Cone Dysfunction in an In Vivo Model of Rod-Cone Dystrophy

**DOI:** 10.3390/ijms21176055

**Published:** 2020-08-22

**Authors:** Mark M. Hassall, Michelle E. McClements, Alun R. Barnard, Maria I. Patrício, Sher A. Aslam, Robert E. Maclaren

**Affiliations:** 1Nuffield Laboratory of Ophthalmology, Department of Clinical Neurosciences, University of Oxford, Oxford OX3 9DU, UK; michelle.mcclements@ndcn.ox.ac.uk (M.E.M.); alun.barnard@eye.ox.ac.uk (A.R.B.); maria.patricio@ndcn.ox.ac.uk (M.I.P.); sheraslam2@yahoo.com (S.A.A.); robert.maclaren@eye.ox.ac.uk (R.E.M.); 2Oxford Eye Hospital, Oxford University Hospitals NHS Foundation Trust, Oxford OX3 9DU, UK

**Keywords:** retina, cone photoreceptors, retinitis pigmentosa, rod-cone dystrophy, gene therapy

## Abstract

Retinitis pigmentosa (RP) is a generic term for a group of genetic diseases characterized by loss of rod and cone photoreceptor cells. Although the genetic causes of RP frequently only affect the rod photoreceptor cells, cone photoreceptors become stressed in the absence of rods and undergo a secondary degeneration. Changes in the gene expression profile of cone photoreceptor cells are likely to occur prior to observable physiological changes. To this end, we sought to achieve greater understanding of the changes in cone photoreceptor cells early in the degeneration process of the *Rho^−/−^* mouse model. To account for gene expression changes attributed to loss of cone photoreceptor cells, we normalized PCR in the remaining number of cones to a cone cell reporter (*OPN1-GFP*). Gene expression profiles of key components involved in the cone phototransduction cascade were correlated with tests of retinal cone function prior to cell loss. A significant downregulation of the photoreceptor transcription factor *Crx* was observed, which preceded a significant downregulation in cone opsin transcripts that coincided with declining cone function. Our data add to the growing understanding of molecular changes that occur prior to cone dysfunction in a model of rod-cone dystrophy. It is of interest that gene supplementation of *CRX* by adeno-associated viral vector delivery prior to cone cell loss did not prevent cone photoreceptor degeneration in this mouse model.

## 1. Introduction

Retinitis pigmentosa (RP) is an incurable inherited retinal disease (IRD) that affects approximately 1 in 4000 people [1] and it most commonly arises from genetic mutations in genes specific to rod photoreceptor cells. It is a rod-cone dystrophy, beginning with rod degeneration and the loss of night vision and peripheral vision. The cone photoreceptor cells then undergo secondary degeneration. In many forms of RP, this secondary cell loss occurs despite the mutated gene not being expressed in cone photoreceptors. The loss of cone photoreceptors is often more debilitating than rod photoreceptor cells loss, as cone photoreceptors are responsible for the central field of vision, high acuity vision and color vision [2,3].

The understanding of cone photoreceptor cell survival and health following the degeneration of rod photoreceptor cells is still being elucidated [4,5]. Studies of mouse models of RP have identified key components of cone photoreceptor cells but their influence on survival is still poorly understood [5,6]. RP can occur as a result of a sporadic mutation or from a variety of inheritance patterns. There are over 200 genes associated with RP (*RetNet* [7]) and a great variety of mutations identified in each gene. The heterogeneity of RP genotypes limits the ability to prepare appropriate gene therapy treatments for all patients. A common intervention approach to preserve cone cell function and prevent cone cell apoptosis would be valuable—a “mutation-independent” strategy. Such a broad treatment could be more widely applied than a narrow therapy targeting only a single mutant variant of RP. Although it would not address the loss of rod photoreceptors, it would potentially offer the opportunity to maintain central visual acuity. In order for this to be a viable gene therapy approach, a greater understanding of the molecular profile of cone photoreceptor cells prior to and during degeneration is critical.

Punzo et al. [8] performed a broad characterization of gene pathway changes in the retinal transcriptome across four mouse models of RP during cone cell death (*Pde6^rd1^*, *Pde6g^−/−^*, *Rho^−/−^* and *Rho*^P23H^). A third of the downregulated genes identified from microarray data were genes involved in metabolism. *Opn1sw* mRNA was downregulated across the different mouse lines, but a cone-specific control for cone cell loss was not included. No papers to date have yet examined changes in cone phototransduction gene expression and correlated those changes with changes in visual or electrophysiological phenotype.

This study measured the changes in expression of genes involved in the cone phototransduction cascade during the period of cone photosensitivity loss in the *Rho^−/−^* mouse model. The *Rho^−/−^* mouse model used in this report also expressed GFP under the control of the *OPN1* promoter in cone photoreceptor cells, providing a control in gene expression profiling that would account for the declining total retinal cone cell count over time in the *Rho^−/−^* mouse model.

This report identifies significant downregulation of the genes *Crx* and *Opn1sw* relative to *OPN1-GFP* expression levels in the *Rho^−/−^* mouse model. These changes occurred prior to detectable changes in cone function and survival. As broad therapeutic approaches are being pursued that might preserve cone photoreceptor cells following primary loss of rod photoreceptor cells, we were interested to identify key regulators that might determine cone cell fate. Since *Crx* acts upstream of *Opn1*, we tested the hypothesis that reversing *Crx* downregulation might delay cone cell death when occurring secondary to rod loss in the *Rho^−/−^* mouse model. This did not however prevent cone cell degeneration, implying that downregulation of cone-specific genes such as *Crx* may be a global secondary effect of cone stress rather than being primary drivers of cone cell degeneration.

## 2. Results

### 2.1. Gene Expression Analysis of Cone Phototransduction Genes Over Time in the Rho^−/−^OPN1-GFP Mouse Model

To provide an understanding of the changes in the expression profile over time for genes involved in the cone phototransduction cascade, tissue samples were collected from age-matched *Rho^−/−^OPN1-GFP* mice and wild-type *OPN1-GFP* mice. Target genes for qPCR analysis were: *Opn1mw*, *Opn1sw*, *Pde6h*, *Crx*, *Cnga3*, *Cngb3* and *Arr3*, as well as the cone *GFP* reporter and *ActB* control genes.

Expression of cone *GFP* did decline from baseline in the *Rho^−/−^OPN1-GFP* model between the PNW2 and PNW25 time points (*p* = 0.043; Appendix A). However, changes in cone *GFP* expression were not significantly different between the *OPN1-GFP* and *Rho^−/−^OPN1-GFP* models (*p* = 0.091) at any time point. Expression of *Rho* was significantly different between the *OPN1-GFP* and *Rho^−/−^OPN1-GFP* models at all timepoints (*p* < 0.001; Appendix A). Detection of all genes were then normalized to levels of the two reference genes *ActB* and *GFP* per sample and then compared to levels in the PNW2 *OPN1-GFP* samples (Figure 1a), as well as *OPN1-GFP* samples at the equivalent PNW time point (Figure 1b), using the 2^−ΔΔ*C*t^ method (Appendix B) [9]. These data comparisons revealed the change in gene expression profile over time for both the wild-type *OPN1-GFP* model and the *Rho^−/−^OPN1-GFP* mouse model. Statistical analyses of the qPCR gene expression data were conducted using a non-linear regression model, full details are available in Appendix A and Appendix A.

*OPN1-GFP* mice did not show significantly reduced expression of any gene over time. Rho*^−/−^OPN1-GFP* mice showed significantly reduced expression of *Crx*, *Opn1sw*, *Opn1mw, Arr3*, *Cnga3* and *Pde6h* genes at PNW25 when compared to the PNW2 baseline. *Crx* declined significantly from baseline from PNW6 onwards (*p* < 0.001). *Opn1sw* and *Cnga3* declined from baseline by PNW17 (*p* < 0.001 and *p* < 0.05 respectively). Expression levels were significantly different between the two mouse models for *Crx* at PNW17 (*p* < 0.001) and PNW25 (*p* < 0.001), *Opn1sw* at PNW17 (*p* < 0.05) and PNW25 (*p* < 0.001), and *Arr3* at PNW25 (*p* < 0.05). The difference between the two genotypes was not significant for *Opn1mw* at any time point. Downregulation of *Crx* in *Rho^−/−^OPN1-GFP* mice showed a seven-fold decline from baseline by PNW 12 and by 30-fold by PNW 17. The *Opn1sw* gene was the most downregulated of the two cone opsins, demonstrating a significant 16-fold reduction in expression by PNW 17 whereas *Opn1mw* declined five-fold by PNW 17 (not significant between genotypes). Analysis of the cone genes are presented in detail in Appendix A.

### 2.2. Characterization of the Rho^−/−^OPN1-GFP Mouse Model Phenotype

The natural history of retinal architecture degeneration was measured using OCT of both eyes in *Rho^−/−^OPN1-GFP* mice prior to sacrifice at timepoints of PNW6, 12, 17, and 25. Figure 2 shows that the outer retinal thickness of *OPN1-GFP* controls varied minimally from baseline and did not change significantly between PNW6 and PNW50 (*p* = 0.40, linear model). Similarly, no differences in outer retinal thickness were evident on the superior-inferior axis at the different time points when measured 750 µm from the nerve head. In *Rho^−/−^OPN1-GFP* mice, the outer retina was significantly different between PNW6 and PNW12 (*p* < 0.001), as well as between PNW12 and PNW17 (*p* < 0.001). The *Rho^−/−^OPN1-GFP* outer retina was unmeasurable on OCT by 17 weeks of age. The rate of thinning was equal for both superior and inferior retina (*p* = 0.36). In *Rho^−/−^OPN1-GFP* mice, after loss of the outer retina, the total retinal thickness at PNW 17 (116.5 ± 10.0 µm) did not thin significantly by PNW 25 (117 ± 17.4 µm), demonstrating that the degeneration detected with OCT was confined to the outer retina. The sensitivity of qPCR to detect remaining rod photoreceptor cells outperforms in vivo OCT imaging. The residual mutant *Rho* transcript detected in Appendix A persists long after the outer nuclear layer of rod photoreceptors is no longer detectable on OCT (Figure 2a).

Given the known loss of retinal structure in the *Rho^−/−^* mouse model [10], for additional information relating to the degeneration of cone photoreceptor cells in this model, we used the *OPN1-GFP* reporter locus to assess the cone cell marker using en face cSLO imaging. Cone cells expressing *OPN1-GFP* are visible as fluorescent dots and manual counting of these correlates with the surviving number of GFP positive photoreceptors [11]. The natural history of cone photoreceptor degeneration was measured using cSLO in *Rho^−/−^OPN1-GFP* mice and *OPN1-GFP* mice at cross-sectional timepoints PNW 6, PNW 12, PNW 17, and PNW 25. Representative images are shown in Figure 3a indicating a loss of GFP positive cone cells in *Rho^−/−^OPN1-GFP* mice over time. Manual counting of the fluorescent dots from standardized images (Figure 3b) revealed that *Rho^−/−^OPN1-GFP* mice began with equivalent GFP positive cells as *OPN1-GFP* controls (*p* = 0.74). *Rho^−/−^OPN1-GFP* mouse cell count then declined over time (−1.1 ± 0.1 cells/day) and approached zero at 25 weeks (34 ± 18). *OPN1-GFP* mice had stable cell counts at PNW6, 12, and 17.

With the loss of GFP positive photoreceptors and loss in outer retinal structure over time, we further tested cone function using a light-adapted ERG protocol [12]. *Rho^−/−^OPN1-GFP* mice were assessed at PMW 6, PNW 12, PNW 17, and PNW 25. The b-wave amplitudes elicited by a range of flash stimulus (−0.5 to 1.5 log cd.s/m^2^) in photopic conditions for these mice are shown in Figure 4a,b. In The cone-specific b-wave of the *Rho^−/−^OPN1-GFP* mice at PNW 6 (137 ± 90 µV) was significantly reduced from baseline (PNW 6) by PNW 17 (6 ± 6 µV, *p* < 0.001) and PNW 25 (3 µV ± 9 µV, *p* < 0.001), but not significantly reduced at PNW 12 (52 ± 14 µV, *p* = 0.12).

For further understanding of the influence of the progressive cone dysfunction in the *Rho^−/−^OPN1-GFP* mouse model over time, head-tracking behavior under photopic conditions was assessed at PNW 6 (*n* = 10 eyes), PNW 10 (*n* = 4 eyes), PNW 12 (*n* = 8 eyes) and PNW 17 (*n* = 10 eyes), Figure 4c. At the 6 week baseline time point, head-tracking of 5.1 ± 2.1 per minute under photopic conditions was significantly different compared to the stationary stimulus null test conditions (0.13 ± 0.08; *p* < 0.001) and was equivalent to responses achieved from mice with no retinal degeneration. Head-tracking behavior did not change significantly from baseline to 10 weeks of age (5.2 ± 1.7, *p* < 0.001) but at 12 weeks of age was reduced significantly to 1.6 ± 1.3 (*p* < 0.01), although this remained significantly higher than the null stimulus condition (*p* = 0.04). The distinction between positive and null test condition head-tracking behavior was lost at 17 weeks (0.8 ± 0.8; 0 ± 0.2; *p* = 0.23).

### 2.3. CRX Gene Supplementation in Rho^−/−^OPN1-GFP Mice

By characterizing the progression of changes in cone function in *Rho^−/−^OPN1-GFP* mouse model, we identified that a downregulation in expression of *Crx* preceded structural and functional changes in cone photoreceptor cells. Gene therapy strategies that aim to preserve and prevent cone loss in rod-cone dystrophies are of great interest and in line with this broad treatment approach, we generated an rAAV.*CRX* construct. This vector was validated and delivered as described in the methods and Appendix A. Mice received injections of rAAV.CRX and in vivo testing at timepoints described in Appendix A. In an attempt to counter the loss of cone photoreceptor cells by early intervention of their gene expression profile, *Rho^−/−^OPN1-GFP* mice were injected at PNW3. Evidence of increased *CRX* mRNA expression above native *Crx* expression was achieved by qPCR Figure 5a, three weeks after subretinal injection of a high dose (1.5 × 10^8^ gc; *n* = 4), low dose rAAV.*CRX* (1.5 × 10^7^ gc; *n* = 4), and PBS sham (*n* = 4). The 2^−ΔΔCt^ method was used to normalize expression to the *ActB* reference gene and against the fellow uninjected eye, before comparing between treatment groups. Treatment with 1.5 × 10^8^ gc of rAAV.CRX showed a significant increase (6.5-fold) in *CRX*/*Crx* gene expression compared to sham injected eyes (1.2-fold; *p* = 0.044) but no significant change in expression was observed in mice treated with 1.5 × 10^7^ gc (*p* = 0.51).

Three antibodies to CRX were tested for species-specific labelling (Abcam, Camrbidge, UK, AB54635; SantaCruz, Dallas, TX, USA, SC-81958; Thermo Fisher Scientific, Waltham, MA, USA, PA5-32182) but none were able to fully distinguish between orthologous mouse and human proteins. Subretinal injections of rAAV.*CRX* (1.5 × 10^8^ gc) caused a marked increase in detectable CRX signal in photoreceptors and cells of the inner nuclear layer Figure 5b,c. A dose-dependent gradient from the site of superior retinal injection was apparent with less CRX signal in the inferior retina, as well as compared to uninjected eyes (Figure 5d–g).

### 2.4. Phenotypic Changes to Rho^−/−^OPN1-GFP Mice Following Subretinal Injection of rAAV.CRX

Assessments of outer retinal thickness revealed that gene supplementation with rAAV.CRX did not slow retinal thinning, Figure 6a. In addition to this, the treatment did not show evidence of in vivo cone cell rescue. There were no significant differences in the number of GFP-positive cells in cSLO retinal images of *Rho^−/−^OPN1-GFP* mice at 1 and 5 weeks post injection. The mean number of GFP-positive cells in sham injected retinas at week 1 post-injection (179 ± 23 cells/area) and week 5 (174 ± 38), were not significantly different from high dose (1.5 × 10^8^ gc AAV.*CRX*) injected retinas at week 1 (132 ± 48) or week 5 (131 ± 41, *p* = 0.96). Nor were sham retinas different from low dose (1.5 × 10^7^ gc AAV.*CRX*) injected retinas at week 1 (134 ± 31) or week 5 (108 ± 39; *p* = 0.76).

Light-adapted ERG assessments were performed to assess cone pathway function three weeks post-injection (PNW6), Figure 6b. At the 25 (log 1.5) cd.s/m^2^ flash stimulus, there were no significant differences in the mean b-wave of uninjected eyes across all groups, demonstrating that the baseline characteristics of each group were similar. The 25 cd.s/m^2^ response of the PBS sham injected eyes did not differ significantly from uninjected fellow eyes at 3 weeks after injection (197 ± 22 µV, 207 ± 13 µV, *p* = 0.53; *n* = 17), demonstrating that the sham injections did not have a significant effect on baseline function. Each group of treated eyes was normalized to fellow uninjected eyes and then compared to the sham injected mice at the 25 cd.s/m^2^ flash stimulus at 3 weeks post-injection using a linear model. The high dose (1.5 × 10^8^ gc) and low dose (1.5 × 10^7^ gc) cohorts showed no significant differences from the sham injected cohort. At 9 weeks post-injection (PNW12), the cone response was largely extinguished in all groups (consistent with the earlier natural history study) and there were no significant differences between injected and uninjected eyes for any group. There were also no significant differences in mean b-wave implicit time at any stimulus intensity between the injected eyes of each group.

Despite no rescue of cone photoreceptor structure or function, OMR testing in photopic conditions were also performed with *Rho^−/−^OPN1-GFP* mice 10 weeks post-injection (PNW 13). The untreated controls showed the expected head tracking behavior (median 4, IQR [3.3, 6.7]) but limited or no head-tracking behavior was recorded in each of the treatment groups (Figure 6c).

## 3. Discussion

This study details the timeline of cone function loss and the declining expression of cone phototransduction transcripts in the *Rho^−/−^OPN1-GFP* mouse. Data presented here show that cone ERG function in *Rho^−/−^OPN1-GFP* mice declines precipitously between PNW6 and PNW12, coinciding with loss of OMR head-tracking behavior between PNW10 and PNW12. The complete loss of cone function precedes the loss of GFP labelled cones demonstrated in vivo imaging by a number of months. Taken together, this suggests that dormant cone photoreceptors exist in the retina of RP mice long after the extinction of photosensitivity, Figure 7. Of interest, we identified that the *Crx* gene was the first of the panel tested to be significantly downregulated. In an attempt to reactivate dormant cone photoreceptors we subsequently delivered *CRX* to these mice but observed no improvements in cone photoreceptor survival or function. It is worth noting that cone photoreceptors constitute such a small proportion of the retinal cell population so that even a true cone rescue effect of a portion of retinal cone cells may not be detectable using these measurements.

The rAAV vector was shown to successfully express the *CRX* transgene within three weeks of subretinal injection. So too, CRX protein expression was demonstrated in the photoreceptors beginning within weeks and persisting to at least 10 weeks beyond injection. However, the inability of rAAV.*CRX* to subsequently rescue or prolong cone photosensitivity raises interesting questions about the role of this gene in the loss of cone function. *CRX* is a transcription factor crucial for the differentiation and maintenance of both rod and cone photoreceptors and its downregulation could reasonably contribute to loss of cone photoreceptor function. Humans with *CRX* mutations demonstrate cone-rod retinal dystrophy [13]. In mice, *Crx* synergistically modulates *Op1sw* and *Opn1sw* expression in cones [14]. Mice with *Crx* mutations have down-regulation of genes in the phototransduction cascade, including *Opn1sw*, *Opn1mw*, and *Arr3* [15]. However, *Crx* is expressed in both cone and rod photoreceptors. The magnitude of *Crx* downregulation observed may be mostly caused by rod photoreceptor death, hence the absence of a cone rescue effect from rAAV.*CRX* injections.

*Arr3* transcript levels appeared lower in *Rho^−/−^OPN1-GFP* mice from PNW6 onwards, but never reached significance. Downregulation of *Arr3* may be an early and sustained response to declining cone photosensitivity, thereby prolonging G-protein coupled phototransduction signaling.

The loss of in vivo cone GFP signal in *Rho^−/−^OPN1-GFP* mice followed a ventral-dorsal pattern (Figure 3a) similar to cone death kinetics shown histologically elsewhere [8]. Notably, the relatively unchanged GFP mRNA expression in Appendix A differed from the clear difference of in vivo GFP protein expression in Figure 3a,d. The differences between transcript and protein detection could be due to the global reduction in protein translation observed in cells undergoing pro-apoptotic stress [16]. The differences may also arise from protein mis-localization and misfolding—and thus loss of GFP fluorescence—observed in cone photoreceptors during degeneration [8].

Opn1mw is expressed throughout the cones of the dorsal and ventral retina, whereas Opn1sw is expressed mainly in the dorsal retinal cones. The significant downregulation of dorsal *Opn1sw* mRNA differed from the unchanged ventral-dorsal *Opn1mw* mRNA levels (Figure 1) and, taken together, deviated from the ventral-dorsal pattern of in vivo cone GFP signal loss (Figure 3a)**.** This difference may reflect differential regulation of the transcriptome in S-cones and M-cones, or alternatively suggests that S-cones may be more vulnerable to cell death in RP than M-cones. The unchanged *Opn1mw* mRNA levels is further evidence that loss of in vivo GFP protein expression in Figure 3a is due to translational mechanisms, not only cone cell loss.

Punzo et al. demonstrated that genes involved in metabolism are greatly down-regulated across multiple mouse models of RP. In particular, insulin/mTOR glucose metabolism pathways are most affected [8]. Similarly, Rod derived Cone Viability Factor (RdCVF) deficiency accelerates cone death in RP rodent models [17], whilst RdCVF supplementation stimulates aerobic glycolysis in cones [18]. The work on insulin/mTOR and RdCVF both converge on the central importance of metabolic starvation in cone photoreceptor death in RP. Both papers identify the high energetic demands of photoreceptor outer segment turnover as early casualties of starving photoreceptors. Our data confirms this negative impact of metabolic starvation on cone opsin transcripts, as well as reduced translation and function of cone proteins such as GFP.

The further characterisation and treatment of upstream common causes—such as metabolic starvation—appear more promising than addressing cone photoreceptor specific transcriptome changes. Developing cone-specific therapies remain important until the wider search for effective genetic and molecular therapies for rod rescue in RP is successful.

## 4. Materials and Methods

### 4.1. Mice and Breeding

In order to study the effect of rod degeneration on the cone phototransduction cascade, the *Rho^−/−^* mouse model of RP also expressed GFP in a subset of cone photoreceptors. This GFP expression provided a control for any potential decline in total retinal cone cell count.

All animals used in this study were treated humanely in accordance with the UK Home Office Regulations and the ARVO statement for the use of animals in ophthalmic and visual research. Animal ethics were approved under UK Home Office project license (PPL) and were also approved by the Oxford animal and welfare review board (PPL 30/2808 from 21 December 2010 to 21 December 2015 and PPL 30/3363 from 21 December 2015 to 21 December 2020). Animals were housed in a 12:12h light-dark cycle. *Rho^tm1Phm^* mice (herein known as *Rho^−/−^* mice) have targeted disruption of the *Rhodopsin* gene, resulting in rod degeneration [10]. Tg(OPN1LW-EGFP)85933Hue mice express enhanced GFP in a subset of cone photoreceptors and are herein known as *OPN1-GFP* mice [19]. Tg(Nrl-EGFP)1Asw mice express enhanced GFP only in the rod photoreceptors and are herein known as *Nrl-GFP* mice [20]. All mice were genotyped using the published PCR protocols or cSLO screening around the age of weaning. *OPN1-GFP* breeders were crossed with *Rho^−/−^* mice to produce F1 progeny with the RP phenotype and GFP expression in the cone photoreceptors (*Rho^−/−^OPN1-GFP* mice). *Nrl-GFP* breeders were crossed with Rho^−/−^ mice to produce F1 progeny with the RP phenotype and GFP expression in rod photoreceptors.

### 4.2. Electroretinography

As a means of assessing cone electrophysiological function during the period of cone photosensitivity loss, electroretinography (ERG) was performed on anaesthetized mice. Mice were positioned on a heated mat in front of the testing console (Colordome Electroretinography machine; Diagnosys LLC, Vision Park, Cambridge, UK) and isolated cone function was tested as described elsewhere [12]. Single-flash stimuli after light adaptation consisted of 25 (1.5 log) cd.s/m^2^; 20 responses were averaged, with an ISI of 1 s.

### 4.3. Optomotor Response Testing

As a method of assessing integrated cone function during the period of cone photosensitivity loss, a custom optomotor system was produced, consisting of a rotating cylinder that allowed rotation speed to be specified. The cylinder was lined with a square-wave grating pattern of vertical black and white stripes, with a spatial frequency of the grating pattern of 0.1 cycles per degree. The test was performed under photopic conditions: the chamber was illuminated from above with a halogen white light source such that the illumination measured at the position of the animal was 2574 lux. To isolate cone responses, mice were light-adapted on the elevated testing platform for five minutes beforehand and then their head-tracking response to the rotating drum testing protocol quantified as described elsewhere [21].

### 4.4. Confocal Scanning Laser Ophthalmoscopy (cSLO) and Optical Coherence Tomography (OCT)

To accurately measure the rod degeneration and any decline in cone photoreceptor cell counts over time, the retinal phenotype was assessed in vivo using confocal scanning laser ophthalmoscopy and spectral domain optical coherence tomography (Spectralis HRA; Heidelberg Engineering, Heidelberg, Germany) as previously described [22]. A standardized region of interest was sampled from the en face fundus images captured with cSLO (Appendix B) and counted manually using the ImageJ cell counter plugin to determine the surviving proportion of GFP-positive cones. Total retinal and outer retinal thickness measurements from the OCT were made manually using calipers on Heidelberg software. Total retinal thickness was measured from the inner margin of the nerve fiber layer to the inner margin of the RPE. Outer retinal thickness was measured from the inner margin of the outer nuclear layer to the inner margin of the RPE.

### 4.5. mRNA Extraction and cDNA Reverse Transcription from Retinal Tissue

To reliably isolate the retinal transcriptome for analysis at the desired timepoints, groups of animals were euthanized at postnatal week (PNW) 2, 6, 12, 17 or 25, the whole neurosensory retina of each eye was harvested by dissection and stored in RNAlater (Thermo Fisher Scientific, Waltham, MA, USA) at −20 °C. Fluorescence-activated cell sorting (FACS) was avoided to prevent cone cell loss and processing-induced changes in the cone transcriptome. RNA was extracted from the retinas using the manufacturer’s instructions for the RNeasy mini kit (QIAgen, Manchester, UK) and the recommended on-column DNase digest with the RNase-Free DNase set (QIAgen, Manchester, UK). The retinal samples were homogenized with a rotor and 30-gauge needle prior to processing. Eluted retinal RNA was reverse transcribed to cDNA using the Superscript III synthesis system (Life Technologies, Paisley, UK) and the included oligo-dT primer. The manufacturer’s instructions were used.

### 4.6. Transcriptome Analysis Using qPCR

To measure the changes in expression of genes involved in cone function in the *Rho^−/−^* mouse model, custom oligonucleotide primers for qPCR were designed to target six genes of the cone phototransduction cascade: *Opn1mws*, *Opn1sws*, *Arr3*, *Pde6h*, *Cnga3*, *Cngb3*; as well as *Crx* (Appendix A). By selecting cone-specific genes, there is no need to remove other cells using FACS, thereby avoiding loss of cells and reducing processing-induced changes in the cone transcriptome. These genes are most relevant because they provide cone photoreceptor function.

Model validation, normalization and reference primer pairs were also designed for *GFP* and *Actb*. Four to six pairs for each gene were first tested and optimized using non-degenerate *OPN1-GFP* mouse retinal cDNA dilution series. The nucleotide sequences of the qPCR primers are provided in Appendix A; from a list of candidates for each gene, the primer pair with 95–103% efficiency was chosen for the test experiments.

All reactions used a commercial qPCR kit (SYBR Green PCR Master Mix; Applied Biosystems, Foster City, CA, USA) and 2 µM final concentration of each forward and reverse primers. All qPCR experiments were performed in triplicate using a commercial real-time PCR machine (CFX Connect Optics Module, BioRad, Hercules, CA, USA). Reactions were performed with the following settings: an initial denaturation step of 95 °C for 10 min, followed by 40 cycles of: 95 °C for 30 s, an annealing temperature of 55 °C and extension at 72 °C for 30 s. Values obtained for the target genes were reported using the 2^−ΔΔ*C*t^ method [9], as detailed in the worked example in Figure A1. Firstly, test gene Ct values were normalized to the geometric mean of the two housekeeping genes, *ActB* and *GFP*, to calculate ΔCt. This provided a double normalization that accounted for variability Secondly, ΔCt values for each mouse type at each timepoint were compared to the *OPN1-GFP* mice at PNW2 to calculate ΔΔCt. Results were interpreted alongside the *GFP* results to account for the decline in cone numbers.

qPCR analysis of transgene expression in rAAV.*CRX* treated mice (Figure 5a) was performed with the same method, using custom oligonucleotide primers outlined in Appendix A.

### 4.7. Recombinant AAV Production and Intraocular Injection

The most down-regulated gene (*CRX)* was cloned into an rAAV vector and over-expressed in the retina of *Rho^−/−^* mice to assess the effect on prolonging or preserving cone photosensitivity in RP. The individual effects of over-expressing *CRX* was explored across two rAAV doses. This study used an early practical time point, postnatal day (PND) 21, for subretinal injection in order to maximize the chance of transgene expression prior to endogenous cone gene downregulation.

rAAV2/2.CAG.CRX.WPRE.pA (AAV.CRX) was produced in HEK293T cells. Cultures were transfected with plasmids containing the expression cassette, RepCap and helper sequences. The HEK293T cells were pelleted, lysed, purified using an Iodixanol gradient, and then purified using an Amicon Ultra 100K filter (Merck, Germany).

The rAAV were validated before in vivo use (Appendix A). In vitro expression of transgenic protein was confirmed by transducing confluent HEK293T cells with rAAV. After culture for three days, immunocytochemistry was performed with primary anti-CRX (PA5-32182, Thermo Fisher Scientific, Waltham, MA, USA) antibody. Secondary staining was performed with Alexa-Fluor 568 donkey anti-rabbit (A10042, Thermo Fisher Scientific, Waltham, MA, USA).

rAAV capsid purity was confirmed with SDS-PAGE. The rAAV concentrate was mixed with protein loading buffer (National Diagnostics, Atlanta, GA, USA), heat denatured and separated electrophoretically on a precast gel (Bio-Rad, Hertfordshire, UK). The capsid protein bands were stained using EZBlue (Sigma-Aldrich, Dorset, UK).

Viruses titre was measured using optimised custom qPCR primers targeting *CRX* (FW-5′-GACAGCAGCAGAAACAGCAG-3′ and RC-5′-GGCTCCAGATGGACACAGTG-3′). rAAV was treated with DNAse I (New England Biolabs, Ipswich, MA, USA) before the capsids were denatured by heating. qPCR was performed as described above and the titre was measured against a known dilution series of pAAV.CAG.*CRX.WPRE.pA* plasmid.

rAAV suspended in phosphate buffered solution (PBS) to give the desired dose in 1.5 µL was delivered by subretinal injection to *Rho^−/−^, OPN1-GFP* mice at PND 21. After the onset of anesthesia, the pupil was dilated. A 6mm circular glass coverslip (VWR International, East Grinstead, UK) was positioned on a viscous coupling gel (Viscotears liquid gel, 0.2 mg/g polyacrylic acid; Alcon Laboratories Ltd., Camberley, UK) and the fundus was visualized using an operating microscope (Leica Biosystems, Wetzlar, Germany).

A Nanofil 10 µL syringe (World Precision Instruments, Sarasota, FL, USA) was assembled with the Nanofil 35G beveled needle (WPI). 1.5 µL of PBS sham or rAAV vector at the chosen titre was delivered by trans-scleral injection into the subretinal space to create a hemiretinal detachment. Antibiotic eye drops (Chloramphenicol 0.5%; Bausch & Lomb, Rochester, NY, USA) and Viscotears were applied topically.

### 4.8. Immunohistochemistry

In order to demonstrate successful retinal transduction, as well as accurately examine the localization of the *CRX* transgene, a subset of injected eyes were collected for immunohistochemistry. Following euthanasia and enucleation, the cornea was excised at the limbus. The zonules were blunt dissected with Vannas scissors and the lens extruded whilst avoiding retinal traction. The eyecups were then transferred to 4% (*w*/*v*) PFA (Thermo Fisher Scientific, Waltham, MA, USA) for 30 min for initial fixation. The eyecups were then sequentially transferred to 10%, 20% and 30% sucrose and then embedded in OCT compound (VWR International, East Grinstead, UK) and stored at −80 °C until sectioning.

Sample blocks were sectioned in 20 µM slices at −22 °C (Cryotome LSE; Thermo Fisher Scientific, Waltham, MA, USA), placed onto polysine-coated glass slides and allowed to dry overnight. All sections were then permeabilized with dilute Triton-X, blocked with donkey serum and incubated with primary anti-CRX (PA5-32182, Thermo Fisher Scientific, Waltham, MA, USA). Secondary staining was performed with Alexa-Fluor 568 donkey anti-rabbit (A10042, Thermo Fisher Scientific, Waltham, MA, USA). The slides were then counter-stained with 4’,6-diamidino-2-phenylindole (DAPI; Invitrogen, Carlsbad, CA, USA). Coverslips were applied with ProLong Diamond Antifade Mountant (Life Technologies, Paisley, UK).

Retinal sections were viewed on a confocal microscope (LSM710; Zeiss, Oberkochen, Germany). Fluorescent cells were located using fluorescence illumination before taking a series of 0.5 µm thickness overlapping XY optical sections. Fluorescence of Hoechst, GFP, and Alexa 568 were sequentially excited and a stack built. Image processing was performed using Volocity (Perkin-Elmer, Waltham, MA, USA) and ImageJ (version 1.43; National Institutes of Health, https://imagej.nih.gov/ij) [23].

### 4.9. Statistical Analysis

Statistical analysis was performed using R software (v3.5.1) [24]. Hierarchical linear and non-linear modelling was performed using the *nlme* package [25,26].

## Figures and Tables

**Figure 1 ijms-21-06055-f001:**
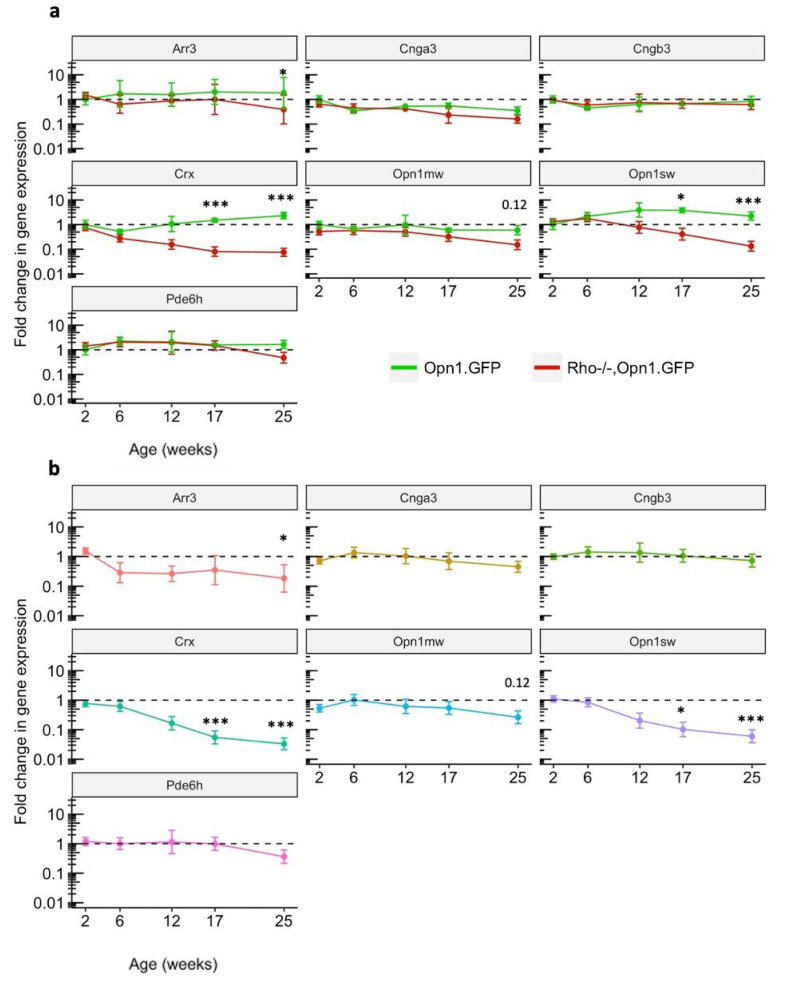
Gene expression analysis of cone phototransduction genes. (**a**) Gene expression data over time (post-natal week, PNW, 2, 6, 12, 17 and 25) showing 2^−∆∆*C*q^ values for *Rho^−/−^OPN1-GFP* mice and *OPN1-GFP* mice. ΔΔCt values calculated by double normalizing the test gene to the mean of *ActB* and *GFP* reference gene levels, then comparing each time point against *OPN1-GFP* mouse expression levels at PNW2 baseline. The dotted line represents the gene expression baseline of *OPN1-GFP* mice at PNW2 for each gene. *Crx* and *Opn1sw* genes show significantly different reduced expression patterns at PNW17 and PNW25, a trend that is already emerging at PNW12. The other genes are not significantly different between genotypes, but many do significantly change from PNW2 baseline. (**b**) The same gene expression data, displaying ΔΔCt values calculated by double normalizing the test gene to the mean of *ActB* and *GFP* reference gene levels, then comparing each time point against *OPN1-GFP* mouse expression levels at the same time point (*OPN1-GFP1* values represented by the dotted line). Full statistical model results available in Appendix A; All values are mean ± SEM. Tissue samples collected at PNW2 (OPN1-GFP *n* = 8, Rho^−/−^OPN1-GFP *n* = 8), PNW6 (*n* = 3, *n* = 5), PNW12, (*n* = 3, *n* = 3), PNW17 (*n* = 3, *n* = 4), and PNW25 (*n* = 3, *n* = 3). * denotes *p* < 0.05, *** denotes *p* < 0.001.

**Figure 2 ijms-21-06055-f002:**
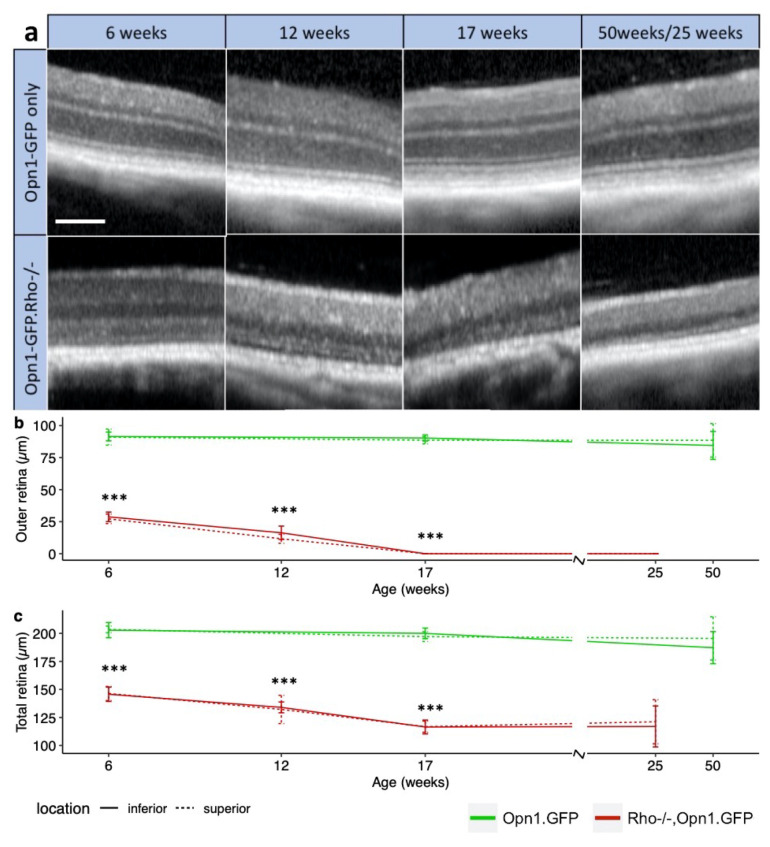
Changes in retinal structural in *Rho^−/−^OPN1-GFP* mice. (**a**) Representative OCT images for Rho*^−/−^*OPN1-GFP mice and OPN1-GFP controls at each time point, showing the loss of outer retina. White scale bar = 100µm. (**b**) Degeneration of the outer retinal thickness and (**c**) total retinal thickness measured on OCT for *Rho^−/−^OPN1-GFP* mice (red) occurs rapidly and differs from non-degenerate *OPN1-GFP* mice (green) as early as PNW6. Total retinal thickness was measured from the inner margin of the nerve fiber layer to the inner margin of the RPE. Outer retinal thickness was measured from the inner margin of the outer nuclear layer to the inner margin of the RPE. ***** denotes *p* < 0.001; values are mean ± SEM. Tissue collected from *Rho^−/−^OPN1-GFP PNW 6 (n* = 7 mice), PNW 12 (*n* = 3), PNW 17 (*n* = 8), and PNW 25 (*n* = 3) and in wild-type *OPN1-GFP* mice at PNW 6 (*n* = 4), PNW 17 (*n* = 6) and PNW 50 (*n* = 2).

**Figure 3 ijms-21-06055-f003:**
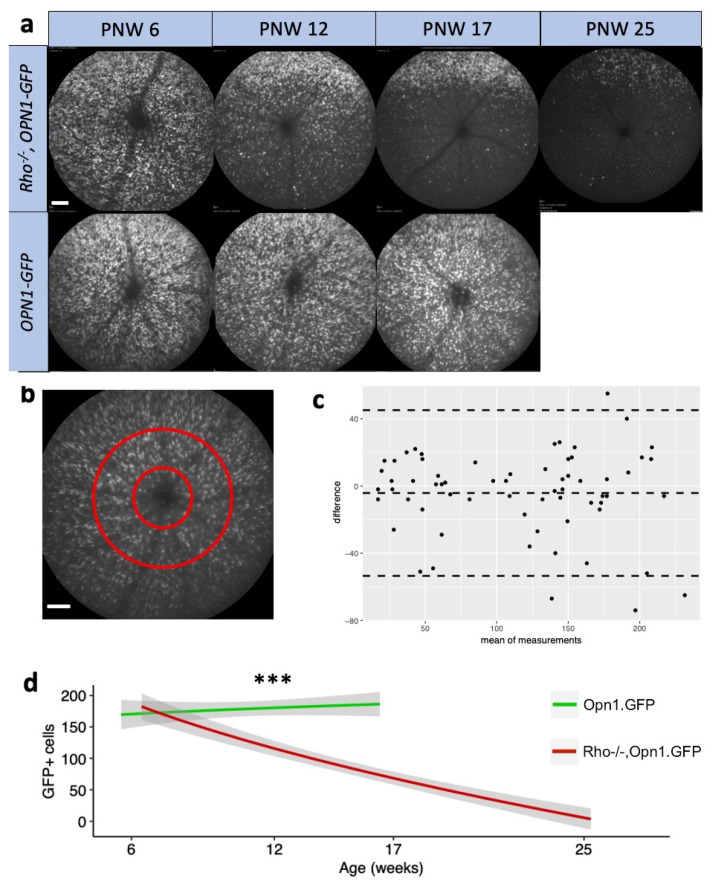
Changes in GFP positive cone photoreceptor cells in *Rho^−/−^OPN1-GFP* mice. (**a**) Representative cSLO images demonstrating that *Rho^−/−^OPN1-GFP* mice show a decline in GFP positive cone cells in vivo, compared to the steady cone population in the *OPN1-GFP* mice. Representative images demonstrate the gross pattern of in vivo GFP positive cone cell loss in *Rho^−/−^OPN1-GFP* mice. White scale bar = 0.1mm (**b**) To quantify the magnitude of loss of cones, a central annulus was sampled for manual particle counts using the ImageJ counter plugin. White scale bar = 0.1mm (**c**) Bland-Altman plot showing the difference in manual counts between the left and right eyes across the mice shows no obvious systematic error. Some mice had substantial inter-eye differences of GFP-positive cone cells, which is similar to the asymmetric progression of IRDs observed in humans. (**d**) Manual cell counts of fluorescent GFP cells in en face cSLO images of the retina showing loss of cone GFP signal in *Rho^−/−^OPN1-GFP* mice and OPN1-GFP mice at cross-sectional timepoints PNW 6 (*n* = 9 mice; *n* = 9 mice respectively), PNW 12 (*n* = 5; *n* = 4), PNW 17 (*n* = 12; *n* = 5), and PNW 25 (*n* = 6; *n* = 0). ***** denotes *p* < 0.001; values are mean cell count ± 95% CI (grey band).

**Figure 4 ijms-21-06055-f004:**
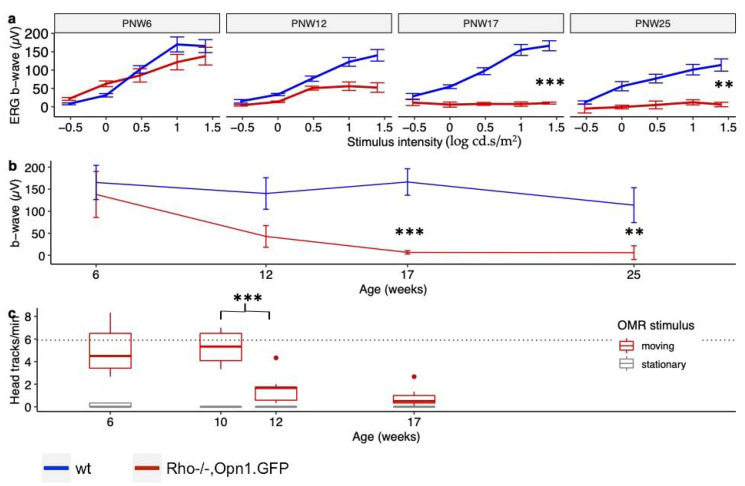
Changes in retinal function in *Rho^−/−^OPN1-GFP* mice. (**a**) Light-adapted ERG b-wave amplitudes for a range of stimulus intensities (−0.5 to 1.5 log cd.s/m^2^) comparing *Rho^−/−^OPN1-GFP* mice and wild-type mice at PNW 6 (*n* = 7; *n* = 6, respectively), PNW 12 (*n* = 2; *n* = 5), PNW 17 (*n* = 7; *n* = 6) and PNW 25 (*n* = 3; *n* = 4). *** denotes *p* < 0.001; ** denotes p < 0.01; All values mean ± SEM. (**b**) The same ERG data, displaying only the light-adapted ERG b-wave amplitudes for 1.5 log cd.s/m^2^ flash stimulus for both mouse strains over time. There is rapid loss of cone function in *Rho^−/−^OPN1-GFP* mice. *All values mean ± SEM.* (**c**) OMR photopic head-tracking behavior in *Rho^−/−^OPN1-GFP* mice at PNW 6 (*n* = 5 mice), PNW 12 (*n* = 3), PNW 17 (*n* = 4), and PNW 25 (*n* = 5). Loss of head tracking occurs between 10 and 12 weeks (*p* < 0.001). Null-stimulus OMR test conditions provided as controls (grey boxes) and ‘wt’ *OPN1-GFP* mouse head-tracking behavior provided as healthy reference (dotted line). All values IQR ± 95% CI, points represent outliers.

**Figure 5 ijms-21-06055-f005:**
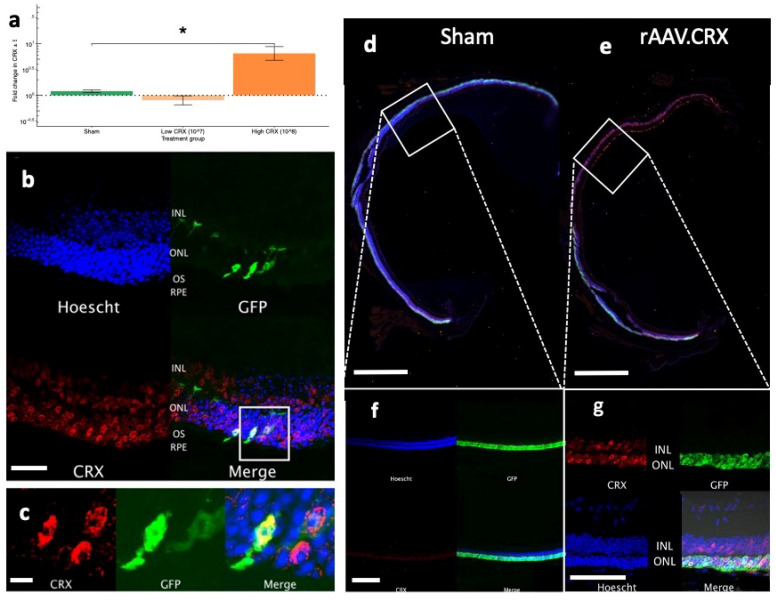
CRX transgene overexpression by subretinal rAAV injections. (**a**) Fold-change in CRX gene expression in *Rho^−/−^OPN1-GFP* mouse retinas injected with high dose rAAV.CRX (1.5 × 10^8^ gc; dark orange; *n* = 4), low dose rAAV.CRX (1.5 × 10^7^ gc; light orange; *n* = 4) or sham PBS (green; *n* = 4). A significant increase occurred in only the 1.5 × 10^8^ gc treated retinas. * denotes *p* < 0.05 compared to sham eyes. All values mean ± SEM. (**b**) Superior retina of an rAAV.CRX (1.5 × 10^8^ gc) treated eye in an *Rho^−/−^, OPN1-GFP* mouse sacrificed at PNW 10 (7 weeks following surgery). Nuclear-specific Crx staining is seen throughout the outer nuclear layer and, more lightly, in the inner nuclear layer. Scale bar = 25µm (**c**) Enlarged area of interest showing colocalization of CRX with endogenous GFP in cone photoreceptors, confined to the nuclei and not extending into the outer segment. CRX staining also occurs in other non-GFP labelled nuclei which are likely rod cells. Scale bar = 5µm (**e**,**g**) rAAV.CRX (1.5 × 10^8^ gc) injected pilot study *Rho^−/−^Nrl-GFP* retinas showing CRX staining in the ONL and INL. rAAV.CRX injected superior retinas showed a far stronger CRX staining pattern compared to the inferior retinas of the injected eyes, as well as contralateral sham injected eyes (**d**,**f**). Scale bars d,e both = 1mm. Scale bars f,g both = 50µm.

**Figure 6 ijms-21-06055-f006:**
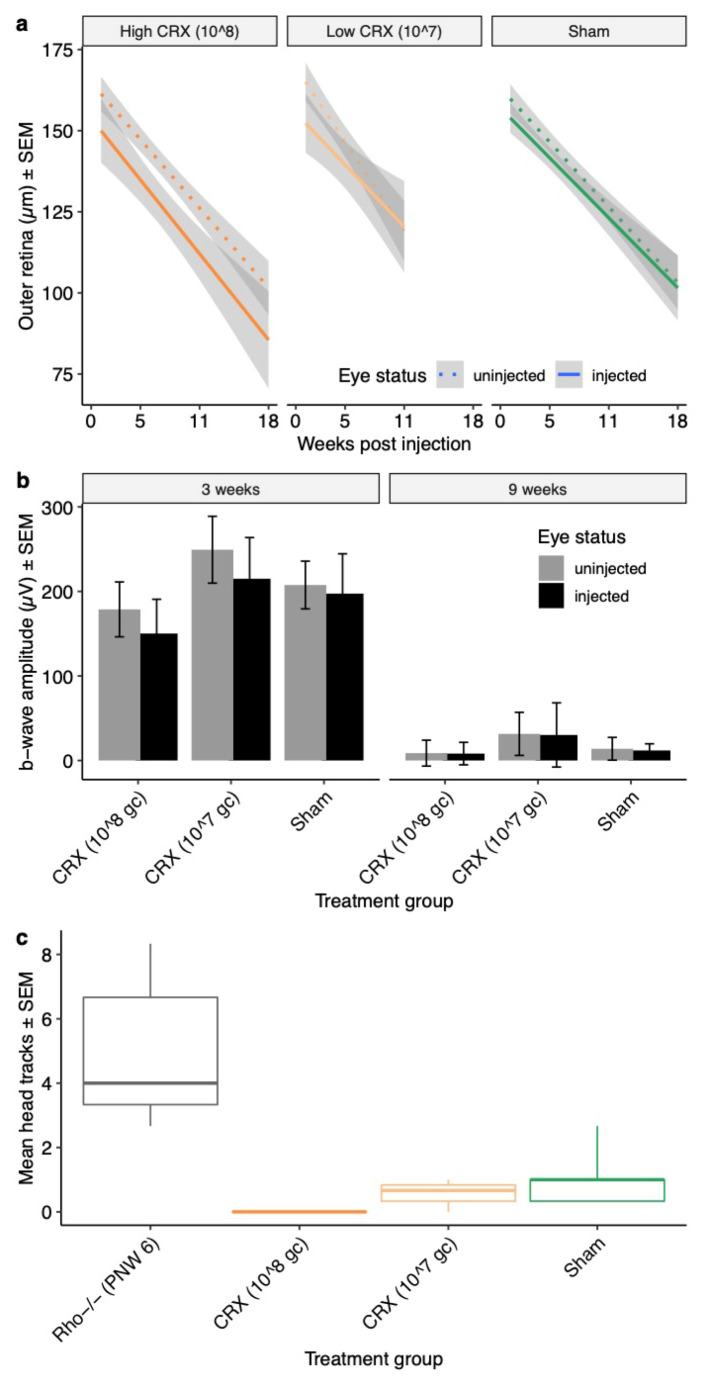
Rescue effect of rAAV.CRX on cone pathway function and retinal structure in *Rho^−/−^, OPN1-GFP* mice. (**a**) Average thickness of superior the outer retina during the study period shows equivalent age-related decline in total retinal thickness of uninjected eyes. No group had slower outer retinal loss in the injected eye. All values are mean µM ± 95% confidence intervals (grey band). (**b**) Light-adapted ERG response to 25 cd.s/m^2^ photopic flash stimulus at PNW 6 (3 weeks post injection) and PNW 12 (9 weeks post injection) in *Rho^−/−^, OPN1-GFP* mice, comparing b-waves of injected and uninjected eyes. None of the groups showed rescue of ERG function in injected eyes at PNW 12. All values are mean µV ± SEM. (**c**) OMR in photopic testing conditions of *Rho^−/−^, OPN1-GFP* mice 10 weeks following rAAV.CRX or PBS sham injection (PNW13). Uninjected *Rho^−/−^, OPN1-GFP* controls at PNW 6 (*n* = 5; grey) are included for comparison. There is limited head-tracking behavior in each of the treatment groups and no rAAV treatment group showed rescue of head-tracking behavior. (All values mean head tracks/minute ± SEM bars. For each panel, treatments groups are high dose AAV.CRX (10^8^ gc/µL; n= 14 mice; dark orange), low dose AAV.CRX (10^7^ gc/µL; n= 17 mice; light orange) or PBS sham (n = 17 mice; green).

**Figure 7 ijms-21-06055-f007:**
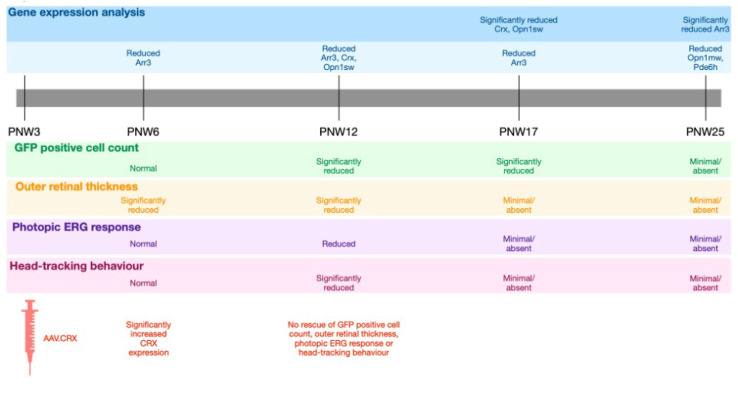
Summary of the changes in gene expression and phenotype over time in *Rho^−/−^, OPN1-GFP* mice.

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
