# Peer review of "Analysis of Early Cone Dysfunction in an In Vivo Model of Rod-Cone Dystrophy"

_ijms, 2020, doi:10.3390/ijms21176055_

Round 1
Reviewer 1 Report
The authors have substantially revised this submission and have adequately addressed the concerns of the original reviewers. The limitations of the study include the fact that the rhodopsin knockout mouse is not good model of retinitis pigmentosa, and the fact that over expression of a regulatory gene such as Crx using gene delivery is a risky proposition at best. Nevertheless, the use of OPN1-GFP allowed them to analyze the transcripts of interest in cone cells that remained viable in the background of degenerating rod photoreceptors, allowing them to conclude that cone photoreceptors persist after the loss of their phototransduction pathway.
Reviewer 2 Report
This is a well-written paper that identifies significant downregulation of the genes Crx and Opn1sw relative to OPN1-GFP expression levels in the Rho-/- mouse model. The experimental protocol is described adequately, and the results and discussion are appropriate.
This manuscript is a resubmission of an earlier submission. The following is a list of the peer review reports and author responses from that submission.
Round 1
Reviewer 1 Report
The authors previously reported that recombinant adeno-associated virus (rAAV) gene therapy was used to deliver enhanced halorhodopsin (eNpHR) to photoreceptors in Pde6brd1 and Rho-/-mice. eNpHR treated eyes performed better in optomotor reflex (OMR) and light-dark box tests. Endogenous human opsin targeted to cone photoreceptor in the early stage of RP may be beneficial to visual function. They measured the changes in expression of genes involved in the cone phototransduction cascade during the period of cone photosensitivity loss in the Rho-/- mouse model. They also constructed rAAV vectors including two of the most down-regulated cloned genes (CRX and OPN1LW) and over-expressed in the Rho-/- mice to access the effect on prolonging or preserving cone photosensitivity in RP.
1. Although only CRX and OPN1LW are judged as down-regulated, Arr3 and Op also look like down-regulated. The reviewer could not understand why the authors chose them.
2. The reviewer agree that dormant cone photoreceptors may be target for reactivation in the subsequent experiments using rAAV gene therapy. However, it is not warranted that to deliver the down-regulated genes CRX and OPN1LW is effective to delay the retinal degeneration
Specific Comments
1. Please unify the description method (ex. line 406, Applied Biosystmens, line 409, BioRad, line 437, Bio-Rad, Hertfordshire, UK, line 438, Sigma-Aldrich, Dorset, UK etc.)
2. line 201-203, Did the location of OPN1LW expression have relation to the location of rAAV injection? Please specify the fundus location where the authors injected the rAAV in methods section (4.7). Superior retina, inferior retina, or post pole?
3. Section 2.3.2, There are not description for figure 3f, 3i, and 3j.
4. Legend of figure 3 and 4, (B), (C), (D) Please change to small letter.
5. Legend of figure 3, (C) and (D) are not in accordance with figure.
Reviewer 2 Report
The authors assessed the changes in cone cells in Rho (-/-) RP mouse model. They found that crx and cone opsin decreased when the cone cell function was decreased. Overexpression of crx and cone opsin by using rAAV was not effective in restoring visual function. While it is worthwhile to know that it is not effective to try to overexpress crx or cone opsin in RP model, this manuscript needs to be improved in several aspects.
Decreased expression of crx and cone opsin would be a part of result from cone cell death, not a cause of the disease. It would be better to evaluate the mechanism of the decreased level of crx and cone opsin, and to try rescue cone cells.